# OpenReview forum: "Detecting Misbehaviors of Large Vision-Language Models by Evidential Uncertainty Quantification"
_ICLR.cc/2026/Conference — ICLR 2026 Poster_

### Official Review · Reviewer_RBSV · 2025-10-29

**Soundness:** 3
**Presentation:** 3
**Contribution:** 3
**Rating:** 6
**Confidence:** 3

**Summary:**

The paper proposes Evidential Uncertainty Quantification (EUQ), a method to detect misbehaviors in Large Vision-Language Models (LVLMs) by quantifying two types of epistemic uncertainty: conflict (CF) and ignorance (IG). Using Dempster-Shafer Theory, EUQ models output-layer features as evidence, enabling efficient detection of hallucinations, jailbreaks, adversarial attacks, and OOD failures in a single forward pass. Extensive experiments on four LVLMs show that EUQ outperforms existing baselines in AUROC and AUPR, with insights into layer-wise uncertainty dynamics.

**Strengths:**

1. Good writing.
2. The proposed method appears to have promising performance.
3. The proposed indicators were effective across different series, validating the effectiveness of the method.

**Weaknesses:**

1. The different types of epistemic uncertainty (CF and IG) quantified are all effective for hallucination detection.
2. There is a lack of baseline data for some hallucination detection in 2025.
3. Some work on the detection of LVLMs using evidence theory has not been discussed fully.
4. It is recommended to test on a larger model, such as 72B, because there are often inconsistencies between small and large models.
5. Can CF and IG complement each other to improve the final detection performance?
6. It's best to separate citations from the main text, for example, using \citep.

**Questions:**

See Weaknesses.

---

> ### Author Response · Authors · 2025-11-22
>
> Thanks for your positive and valuable feedback. We made efforts to address your every concern and question. If we have any misunderstandings or further questions, please feel free to let us know and we will reply quickly.\
> **Q1: Effectiveness of both CF and IG for hallucination detection (Weakness 1)**
> >**Reply**: Hallucinations may stem from multiple and heterogeneous sources. Our method quantifies epistemic uncertainty along two complementary dimensions, namely conflict (CF) and information gap (IG), and both have proven effective for hallucination detection. These results suggest that hallucinations may be associated either with conflicting feature-level evidence or with missing or incomplete information.
>
> **Q2: Experiments on 72B models and 2025 data (Weakness 2 & Weakness 4)**
> >**Reply**: Following the comment, we have added additional baseline data for hallucination detection using the dataset introduced in the CVPR~2025 work [1].
> We also conducted experiments on Qwen2.5-VL-72B-Instruct [2] to evaluate performance on a larger model.
> The results indicate consistent behavior of our method and the baselines on the 2025 dataset and the 72B model.
> The experimental results on this dataset are shown in **Table 1**. We will add the complete results for the remaining three misbehavior types, together with analysis, in the revised version.
>
> **Table 1: Comparison of our method and baselines on Qwen2.5-VL-72B using the merged dataset of POPE [3], R-bench [4], and [1].**
> |-|AUROC|AUPR|
> |-|-|-|
> |SC|0.701|0.874|
> |SE|0.609|0.856|
> |PE|0.783|0.558|
> |Ln-PE|0.783|0.553|
> |CF|**0.817**|**0.884**|
> |IG|0.763|0.872|
>
> **Q3: Discussion of works on DST-based LVLM detection (Weakness 3)**
> >**Reply**:
> Existing research on applying evidence theory to LVLMs (e.g., [5,6]) follow a paradigm distinctly different from ours. These methods typically rely on evidential deep learning (EDL), which is based on Subjective Logic (SL), to improve model performance while producing uncertainty estimates.
> These approaches require explicit training or fine-tuning, even for large models, making them relatively costly and less scalable.
> In contrast, our method focuses on detecting misbehaviors. EUQ operates directly on pretrained models without any additional training and leverages the full expressive capacity of DST rather than the SL formulation. We hope this work broadens the perspective on DST-based methods and highlights an alternative evidential approach for deep learning models and LVLMs.\
> Although DST-based literature is relatively limited, we aim to discuss all key explorations. We would be grateful if the reviewer could suggest specific works, which we will incorporate into the revised version.
>
> **Q4:  Complementarity of CF and IG in detection (Weakness 5)**
> >**Reply**:
> We agree that the complementarity of CF and IG deserves investigation and tested two straightforward fusion strategies for hallucination detection:
> >- **Conjunctive rule (&)**: flag if both CF and IG exceed their respective thresholds;
> >- **Disjunctive rule (&#124;)**: flag if either CF or IG exceed their respective thresholds.
> >
> >As shown in **Table 2**, combining CF and IG can leverage complementary uncertainty signals, improving hallucination detection performance. The experimental setup is consistent with Table 1, and the threshold for each method was determined based on its respective Youden index.
>
> **Table 2: Performance comparison of CF, IG, and their fusion strategies for hallucination detection.**
> |-|Accuracy|Precision|Recall|F1 Score|
> |-|-|-|-|-|
> |CF|0.851|**0.929**|0.885|0.907|
> |IG|0.859|0.921|0.906|0.914|
> |CF&IG|0.835|**0.929**|0.866|0.896|
> |CF&#124;IG|**0.873**|0.922|**0.924**|**0.923**|
>
>
> **Q5:  Typo (Weakness 6)**
> >**Reply**: We thank the reviewer and will adjust citations to use \citep appropriately.
>
>
>
> **Reference**
> >[1]  Liu J, et al. PhD: A ChatGPT-Prompted Visual Hallucination Evaluation Dataset[C]//CVPR 2025.\
> [2] Bai S, et al. Qwen2. 5-vl technical report[J]//arXiv 2025.\
> [3] Li Y, et al. Evaluating Object Hallucination in Large Vision-Language Models[C]//EMNLP 2023.\
> [4] Wu M, et al. Evaluating and Analyzing Relationship Hallucinations in Large Vision-Language Models[C]//ICML 2024.\
> [5] Li Y, Rügamer D, et al. Calibrating LLMs with Information-Theoretic Evidential Deep Learning[C]//ICLR 2025.\
> [6] Ma H, et al. Estimating LLM Uncertainty with Evidence[J]//arXiv 2025.

---

> ### Author Response · Authors · 2025-12-01
> **Summary of Rebuttal for Reviewer RBSV**
>
> # Summary of Rebuttal for Reviewer RBSV
>
> 1. **Effectiveness of CF and IG for hallucination detection**
> We clarified that CF and IG capture two complementary forms of epistemic uncertainty, and both are effective for hallucination detection. The updated results confirm their utility.
>
> 2. **Experiments on 72B models and 2025 datasets**
> **We added experiments on the CVPR 2025 hallucination dataset and Qwen2.5-VL-72B-Instruct.** The results are consistent with our original findings. Results for the remaining misbehavior types have also been included in the revised version.
>
> 3. **Discussion of DST-based LVLM detection**
> We updated the related-work discussion to distinguish our DST-based detection method from SL/EDL-based approaches that require training or fine-tuning. **All relevant DST-based works have been added to the revision.**
>
> 4. **Complementarity of CF and IG**
> We evaluated two fusion rules (conjunctive and disjunctive). The results show that combining CF and IG provides complementary gains and improves hallucination detection.
>
> Although Reviewer RBSV has not responded, we believe these updates adequately address all concerns.

---

### Official Review · Reviewer_kLDs · 2025-10-31

**Soundness:** 2
**Presentation:** 2
**Contribution:** 2
**Rating:** 4
**Confidence:** 4

**Summary:**

This paper proposes Evidential Uncertainty Quantification (EUQ), a method for detecting misbehaviors in LVLMs including hallucinations, jailbreaks, adversarial vulnerabilities, and out-of-distribution failures. The authors also investigate the conflict and ignorance uncertainty, and argue these two forms of epistemic uncertainty are sources for the misbehaviors.

**Strengths:**

Interesting paper to read as it classifies different types of misbehaviors in VLMs and it is observed that CF/IG can be used to distinguish different types of misbehaviors in VLMs.

**Weaknesses:**

1. Although Figure 4 and the appendix visualizations distinguish misbehavior types, there is no deeper linguistic or visual semantic analysis explaining why certain errors yield high CF or IG.
2. Thresholding (which could vary across LVLMs, datasets, or misbehavior categories) would have to be determined externally. Additionally, since the authors propose metrics to evaluate misbehaviors in VLMs and make observations, the size of the datasets and the chosen VLMs (four VLMs with ≤ 8B parameters) appear somewhat small. What model is evaluated in Figure 5 for model size analysis is not clear.
3. I also think this paper would benefit from including more baselines for hallucination/jailbreak detection. Predictive entropy is a quite simple baseline.

Minor: typos on line 428 for 'adn'

**Questions:**

See weakness.
1. Can you provide more linguistic or visual semantic analysis explaining why certain misbehaviors yield high CF or IG?
2. Can you compare IG and CF with more baselines besides SE (e.g., POPE, HiddenDetect...) / add more datasets/LVLMs to validate the generalization ability of these two metrics and the validity of the observations?

[Evaluating Object Hallucination in Large Vision-Language Models](https://aclanthology.org/2023.emnlp-main.20/) (Li et al., EMNLP 2023)

[HiddenDetect: Detecting Jailbreak Attacks against Multimodal Large Language Models via Monitoring Hidden States](https://aclanthology.org/2025.acl-long.724/) (Jiang et al., ACL 2025)

---

> ### Author Response · Authors · 2025-11-22
>
> Thanks for your positive and valuable feedback. We made efforts to address your every concern and question. If we have any misunderstandings or further questions, please feel free to let us know and we will reply quickly.\
> **Q1: Explanation of Figure 4 (Weakness 1 & Question 1)**
> >**Reply**: The primary goal of this visualization is to demonstrate a key distinction between our proposed Conflict (CF), Ignorance (IG), and predictive entropy (PE), highlighting our key observation (Figure 1 in the main paper) that LVLM misbehaviors could stem from distinct sources.
> To evaluate the effectiveness and discriminative power of our method from the perspective of the distribution of uncertainty measures, we analyze CF, IG, and predictive entropy (PE) across different types of misbehaviors. As shown in Figure 4, for example, OOD samples show a clearer separation in IG than in CF, suggesting that the model is less familiar with these examples and has limited relevant knowledge. This indicates that different error types correspond to different patterns in the model’s internal representations, which PE alone fails to capture. Understanding the model’s complex internal behaviors requires dedicated interpretability studies, which we consider a valuable direction for future work.
>
> **Q2: Clarification of thresholding, datasets and models (Weakness2)**
> >**Datasets and models**:\
> Our work is motivated by a unified view of LVLM misbehaviors: different error types arise from different sources of uncertainty. To rigorously evaluate this perspective, our experiments are explicitly designed to cover multiple distinguished misbehavior categories, rather than focusing on a single one.
> Specifically, we include:
> > - **Four distinguished  misbehavior scenarios**\
> >   Hallucination, Jailbreak, Adversarial, and OOD.
> >
> > - **A total of 11 datasets**\
> > POPE, R-Bench, FigStep, Hades, VisualAdv, typographic attacks, ANDA, PGN, MM-DecodingTrust, MNIST, GTSRB.
> >
> > - **29,000 test instances**
> >    6,500 in the main text and 22,630 in the appendix.
> >
> > - **Four LVLMs**\
> >DeepSeek-VL2, Qwen2.5-VL, InternVL2.5, MoF-Models.
> >
> >We further observed that the trends remain consistent across architectures.
> >Our ablation studies cover models up to 38B parameters. As shown in **Table 1**, we further evaluate hallucination detection on Qwen2.5-VL-72B-Instruct, and the results remain consistent with those reported in Table 4 (lines 2398–2399) in the main text. To further strengthen our observations, we plan to include evaluations on larger models in the revised version.
> >We acknowledge the oversight in not clearly specifying the models used in Figure 5, which were from the InternVL family. This will be clearly stated in the revised version.\
> >Notably, other reviewers regarded our experiments as *extensive and covering lots of* different settings, highlighting this as a merit of our work.
>
> **Table 1: Comparison of our method with baselines on Qwen2.5-VL-72B using the merged dataset of POPE, R-Bench, and [1], as recommended by Reviewer RBSV.**
> |-|AUROC|AUPR|
> |-|-|-|
> |SC|0.701|0.874|
> |SE|0.609|0.856|
> |PE|0.783|0.558|
> |Ln-PE|0.783|0.553|
> |CF|**0.817**|**0.884**|
> |IG|0.763|0.872|
>
> >**Thresholding**\
> Briefly, ROUGE-L measures similarity between generated and reference texts. In our evaluation, we compute the ROUGE-L score between the model’s response (e.g., “The answer is A. (option)”) and the correct option (“A”); if the score exceeds a threshold of 0.5, the answer is considered correct. This threshold is a widely adopted convention in prior works [2,3]. This procedure is agnostic to the model, dataset, or type of misbehavior.
>
> **Reference**
> >[1] Liu J, et al. PhD: A ChatGPT-Prompted Visual Hallucination Evaluation Dataset[C]//CVPR 2025.\
> [2] Kuhn L, Gal Y, Farquhar S. Semantic Uncertainty: Linguistic Invariances for Uncertainty Estimation in Natural Language Generation[C]//ICLR 2023.\
> [3] Duan J, et al. Shifting attention to relevance: Towards the predictive uncertainty quantification of free-form large language models[C]//ACL 2024.

---

> ### Author Response · Authors · 2025-11-22
>
> **Q3: Clarification of baselines(Weakness 3 & Question 2)**
> >**Reply**: Our work focuses on general misbehavior detection, which encompasses four distinct types of misbehaviors (e.g., hallucination, jailbreak, adversarial vulnerabilities, failures to generalize out-of-distribution). In real-world settings, the specific type of misbehavior is often unknown in advance. Therefore, we prioritized general uncertainty-based baselines that are applicable across multiple failure types, rather than those designed for a single specific misbehavior.
> Regarding the mentioned references:
> The work [1] focuses on evaluating object hallucination rather than providing a detection method, making it less directly comparable.
> We appreciate the reference to [2]. As shown in **Table 2**, we have added HiddenDetect as an additional baseline and will include it in the revised version, as suggested.
> Its performance is not particularly strong in our experiments, likely because HiddenDetect heavily relies on the presence of explicit refusal cues (e.g., “sorry”). In our prompt design, the model is compelled to select an option, limiting such cues and thus affecting HiddenDetect’s effectiveness.
>
> **Table 2: Comparison of HiddenDetect with our CF and IG on MoF across all misbehavior datasets.**
> |Dataset|Metric|HiddenDetect|CF|IG|
> |-|-|-|-|-|
> |Hallucination|AUROC|0.828|**0.855**|0.553|
> |-|AUPR|**0.845**|0.838|0.757|
> |Jailbreak|AUROC|0.699|**0.886**|0.859|
> |-|AUPR|0.586|0.534|**0.860**|
> |Adversarial|AUROC|0.592|0.868|**0.999**|
> |-|AUPR|**0.887**|0.832|0.856|
> |OOD|AUROC|0.594|0.979|**0.999**|
> |-|AUPR|0.778|**0.930**|0.855|
> |Average|AUROC|0.678|**0.897**|0.853|
> |-|AUPR|0.774|0.784|**0.832**|
>
>
> **Reference**
> >[1] Li Y, et al. Evaluating Object Hallucination in Large Vision-Language Models[C]//EMNLP 2023.\
> [2] Jiang Y, et al. HiddenDetect: Detecting jailbreak attacks against multimodal large language models via monitoring hidden states[C]//ACL 2025.

---

> > ### Author Response · Authors · 2025-12-01
> > **Summary of Rebuttal for Reviewer kLDs**
> >
> > # Summary of Rebuttal for Reviewer kLDs
> >
> > 1. **Explanation of Figure 4**\
> > We clarified that Figure 4 illustrates the distinction between Conflict (CF), Ignorance (IG)**, and predictive entropy (PE), aligning with our main observation that LVLM misbehaviors stem from different uncertainty sources. We showed that CF and IG exhibit distinct patterns. We also noted that deeper understanding of these behaviors requires further interpretability analyses.
> >
> >
> > 2. **Clarification of thresholding, datasets, and models**\
> > We clarified our experimental design, which intentionally covers **four misbehavior scenarios, 11 datasets, 29,000 test samples, and four LVLM architectures**, a breadth that other reviewers also highlighted positively. We added additional evaluations on **Qwen2.5-VL-72B**, with consistent trends. For correctness evaluation, we follow the standard ROUGE-L > 0.5 threshold widely adopted in prior work.
> >
> > 3. **Clarification of baselines**\
> > We explained that our goal is general misbehavior detection, requiring baselines that apply across diverse failure types rather than specialized ones. **We added HiddenDetect as suggested**, and the newly included results show that it performs less effectively in our setting because it relies on refusal cues that do not appear in forced-choice prompts. All corresponding updates have been incorporated into the revised version.
> >
> >
> > Although Reviewer kLDs has not yet responded, we believe these updates adequately address the concerns raised.

---

### Official Review · Reviewer_TyT1 · 2025-11-01

**Soundness:** 4
**Presentation:** 4
**Contribution:** 4
**Rating:** 8
**Confidence:** 2

**Summary:**

The authors present a training-free method to interpret the uncertainty of a VLM by computing the per-token conflict and ignorance uncertainty in the model's predictions. The authors show that their method outperforms other uncertainty interpretation methods with moderate compute overhead. The paper's methods are evaluated across hallucination, jailbreaking, and out-of-distribution generalization tasks at different model scales.

**Strengths:**

The main strengths of the paper are:
1) Disaggregating the uncertainty into conflict and ignorance uncertainty to interpret the uncertainty of a VLM. This allows the authors to measure uncertainty in different contexts (e.g. hallucination, jailbreaking, out-of-distribution generalization).
2) The experiments are thorough and conducted on multiple model families and expressly evaluated at many scales.

**Weaknesses:**

There are no major weaknesses in the paper. However, in Figure 1 in the paper, the authors show an illustrative example of measuring uncertainty in chain-of-thought reasoning. It would be useful to see examples of how the authors' proposed method can identify uncertainty in these reasoning traces. Currently the authors only evaluate their method on benchmarks which often only measure uncertainty on shorter token sequences.

**Questions:**

1) The paper does not explicitly introduce any mechanism to focus uncertainty on key or semantically important tokens. Would this make it unsuitable for tasks such as dense image captioning or long reasoning traces where only some tokens are key to measuring uncertainty?

---

> ### Author Response · Authors · 2025-11-22
>
> Thanks for your positive and valuable feedback. We made efforts to address your every concern and question. If we have any misunderstandings or further questions, please feel free to let us know and we will reply quickly.\
> **Q1: CoT Uncertainty Demonstration**
> >**Reply**:
> We appreciate your positive assessment of our work and the constructive comments. In our initial experiments, we measured uncertainty on shorter token sequences to enable efficient large-scale assessment of the proposed method. We fully agree that characterizing uncertainty in free-form outputs of LVLM, specifically within CoT reasoning, will be more practically valuable.
> Following your insightful suggestion, we calculated the CF (top example) and IG (below example) for the CoT example in Figure 1 of the paper to identify uncertainty in these reasoning traces.\
> The reasoning steps of the example “Goldfish caused by conflict between text and background image” are as follows. We bold the 10 words with the highest **CF** values, and italicize the top 3 among them:
> >1. I first note the text stating “answer: ***diaper(top1)***.”
> 2. I then observe a **goldfish** in the **background**.
> 3. This is an **unusual design** for baby **products**.
> 4. Although a **goldfish** print on a ***diaper(top3)*** is uncommon and casts **doubt**, it can be reasonable.
> 5. The answer is ***diaper(top2)***.
>
> >Similarly, for the example “Paraglider caused by Ignorance due to missing information,” we highlight the words based on their **IG** values as follows:
> >1. I see **a curved** object with orange-yellow stripes.
> 2. ***Honestly(top2)***, I can't **immediately** identify it; it could be part **of** a toy or some kind of decoration.
> 3. The color pattern ***vaguely(top1)*** **resembles** a **rainbow**.
> 4.  I'm guessing this might be a "rainbow toy."
> 5. The answer **is** ***rainbow(top3)*** toy.
>
> >We observe that CF primarily focuses on concrete objects (e.g., diaper), whereas IG captures not only objects but also modifiers (e.g., vaguely). Moreover, the tokens corresponding to the final decision in the CoT are among the top 3 CF/IG values, indicating the model’s uncertainty in its decision. In the revised version, we will include text heatmap visualizations to visually demonstrate this uncertainty.
>
> **Q2: Focusing on uncertainty of key tokens**
> >**Reply**:
> Thank you for this thoughtful suggestion. We have considered approaches that weight tokens based on their importance, such as SAR [1]. However, these methods typically require additional resources to identify important tokens, often relying on external models, which greatly increase per-sample latency overhead. In contrast, our method is lightweight and plug-and-play for misbehavior detection. Exploring explicit token importance mechanisms remains a valuable direction for future work.
>
> **Reference**
> >[1] Duan J, Cheng H, Wang S, et al. Shifting attention to relevance: Towards the predictive uncertainty quantification of free-form large language models[C]//ACL 2024.

---

> > ### Author Response · Authors · 2025-12-01
> > **Summary of Rebuttal for Reviewer TyT1**
> >
> > # Summary of Rebuttal for Reviewer TyT1
> >
> > 1. **CoT Uncertainty Demonstration**\
> > In response to the reviewer’s suggestions, we conducted additional analyses to better illustrate how our proposed uncertainty scores operate within free-form CoT reasoning. Specifically, we computed CF and IG on the CoT examples in Figure 1 and highlighted the most uncertain tokens within the reasoning steps. The results show that CF tends to capture uncertainty around concrete objects, while IG also emphasizes modifiers expressing hesitation (e.g., “vaguely”). These findings indicate that our method effectively identifies uncertainty in the model’s reasoning process. **We have included these visualizations in the revised version.**
> >
> >
> > 2. **Focusing on uncertainty of key tokens**\
> > Regarding the reviewer’s suggestion on token-importance weighting, existing approaches typically require external models or additional computations that increase latency, whereas our method remains lightweight and plug-and-play. Exploring explicit token-importance mechanisms is a promising direction for future work.
> >
> > Although Reviewer TyT1 has not provided a response, we believe these updates adequately address the concerns raised.

---

### Official Review · Reviewer_iGLE · 2025-11-04

**Soundness:** 3
**Presentation:** 3
**Contribution:** 3
**Rating:** 4
**Confidence:** 2

**Summary:**

The paper studies detecting the misbehaviors of VLMs, in particular, adversarial inputs, Jailbreak inpus, OOD inputs, and model hallucination. The method then extracts the logits from the VLM and process it using "Evidence Theoryt" and get an uncertainty quantification score. The authors then utilize this score to perform misbehavior detection and observe improved performance compared with baseline approaches on. a variety of methods.

I have to say I really cannot understand the method. The paper's technique involves too many concepts I have never heard about:

- Dempster-Shafer Theory
- Aasic belief assignment, BBA
- Degree of conflict
- conflict (CF) and ignorance (IG)
- Least Commitment Principle (LCP)

I don't think this is the author's problem since this is all due to my lack of knowledge. However, my reviews may not provide lots of useful inputs. I would also appreciate it if the authors could explain things in simpler language, e.g. through an algorithm block, such that I could understand how things are implemented in practice.

**Strengths:**

- The method seems to be very mathematically rigorous

- The paper considered lots of datasets and models.

**Weaknesses:**

- The proposed method does not seem to have too much novelty; I can't see why the proposed method is specific to VLM or why it cannot be applied on e.g. BERT, ResNet, LLM.

- It is unclear if the method is applicable to closed-source model since the method requires access to logits.

- The choice of baseline is little bit confusing, semantic entropy is for uncertainty quantification over free form generation, but many tasks here only require a single word as the output (if I understand correctly). What is the point of performing clustering here?

**Questions:**

- Why is the column title near lines 393 and 394 "Method" rather than "model"?

- "For multiple-choice and yes/no tasks, correctness is assessed using ROUGE-L Lin (2004) (threshold > 0.5)." I don't understand why ROUGE is needed here; isn't accuracy applicable?

- The prompt forces the model to hallucinate (line 1229)
```
Please check whether the following description matches
the picture content. Just answer yes or no without explanation.
<image caption>,
```
Has the author tried providing the model with the option to reject? Some recent work (e.g. https://arxiv.org/abs/2505.11804) shows that if prompted properly.

---

> ### Author Response · Authors · 2025-11-22
>
> Thanks for your positive and valuable feedback. We made efforts to address your every concern and question. If we have any misunderstandings or further questions, please feel free to let us know and we will reply quickly.\
> ***Clarification of the method and key concepts***
> >We agree that having a clear view of the underlying theoretical foundations is important for understanding our motivation and method. Accordingly we provide detailed explanations of these concepts in the appendix, and for a deeper understanding, the original source cited in the main text can be consulted: *Shafer, G., A Mathematical Theory of Evidence, Princeton University Press, 1976.*\
> We then illustrate these concepts with a concrete example.
>
> **Dempster shafer theory and basic belief assignment**
> >**Dempster shafer theory (DST)** extends classical probability theory to more flexibly represent uncertainty.
> In classical probability, a probability mass function allocates belief (i.e., probability) only to individual hypotheses.
> DST instead uses a basic belief assignment (BBA), which can assign belief to any subset of the hypothesis space.
> Thus, a BBA plays the same foundational role in DST as a probability mass function does in probability theory, while providing greater expressive power for representing uncertainty.
> Consider a light-bulb switch whose state can be either $\textbf{On}$ or $\textbf{Off}$, i.e., the hypothesis space $\mathcal {H}=${$\{\text{On},\,\text{Off}\}$}.
> In classical probability, probabilities are assigned only to the individual states:
> $$P(\text{On}) + P(\text{Off}) = 1.$$
> If $P(\text{On}) = P(\text{Off}) = 0.5$, the model cannot tell whether this means that the bulb is truly equally likely to be $\textbf{On}$ or $\textbf{Off}$, or simply that we do not know its state.
> DST addresses this limitation by introducing a **Basic Belief Assignment (BBA)**: $m(\cdot)$ defined over the power set $2^{\mathcal {H}}$:
> $$\sum_{S\subseteq \mathcal {H}} m(S)=1, \quad m(\emptyset)=0.$$
> Here, $S$ may be a single state (e.g., $\{\text{On}\}$) or the full set $\mathcal {H}=${$\textbf{On}, \textbf{Off}$}.
> This allows for $2^{|\mathcal {H}|} - 1 = 3$ distinct mass assignments, enabling richer uncertainty representation.
> Importantly, the belief assigned to the full set directly quantifies *ignorance*, i.e., how much uncertainty we have about whether the bulb is $\textbf{On}$ or $\textbf{Off}$.
>
> **Example: conflict and ignorance from three observers**
> >Consider again the frame $\mathcal {H}=\{\text{On},\text{Off}\}$.
> Three independent observers provide BBAs describing the state of the light bulb:
> $$m_1(\{\text{On}\})=0.3,\quad
> m_1(\{\text{Off}\})=0.2,\quad
> m_1(\mathcal {H})=0.5,$$
> $$m_2(\{\text{On}\})=0.6,\quad
> m_2(\{\text{Off}\})=0.2,\quad
> m_2(\mathcal {H})=0.2,$$
> $$m_3(\{\text{On}\})=0.1,\quad
> m_3(\{\text{Off}\})=0.8,\quad
> m_3(\mathcal {H})=0.1,$$
> The belief each observer assigns to the full set $\mathcal {H}$ quantifies **ignorance**:
> $$IG_1 = m_1(\mathcal {H})=0.5,\quad IG_2 = m_2(\mathcal {H})=0.2,\quad IG_3 = m_3(\mathcal {H})=0.1.$$
> To combine two sources of evidence, DST uses *Dempster's rule of combination*, which is formally expressed as
> $$K = \sum_{B \cap C = \emptyset} m_1(B) \, m_2(C),$$
> where the sum is over all pairs of mutually exclusive subsets $B$ and $C$.
> Here, $K$ quantifies the **conflict** between the two BBAs.
> For this example:
> $$K_{12} = m_1(\{\text{On}\})\, m_2(\{\text{Off}\}) + m_1(\{\text{Off}\})\, m_2(\{\text{On}\}) = 0.3\cdot 0.2 + 0.2\cdot 0.6 = 0.18.$$
> $$K_{23} = m_2(\{\text{On}\})\, m_3(\{\text{Off}\}) + m_2(\{\text{Off}\})\, m_3(\{\text{On}\}) = 0.6\cdot 0.8 + 0.2\cdot 0.1 = 0.5.$$
> As shown, $K_{23} > K_{12}$, indicating that $m_2$ and $m_3$ exhibit stronger disagreement than $m_1$ and $m_2$, resulting in a higher conflict value.
>
> **Least commitment principle (LCP)**
> >To obtain more reliable evidence weights, we adopt LCP to estimate the parameters $A$ and $A$ directly from the model’s features and weights. This enables EUQ to remain fully training-free.
> The Least Commitment Principle (LCP) is a fundamental guideline in DST, serving the same purpose as the maximum entropy principle in probability theory: *do not assume more than what the evidence actually supports.*

---

> > ### Author Response · Authors · 2025-11-22
> > **EUQ algorithm block**
> >
> > **EUQ algorithm block**
> > >**Inputs.**
> > Projection input: ${Z}=(z_1,\dots,z_I)\in{R}^{I}$
> > Projection weights and bias: ${W}\in{R}^{I\times J},\;{b}\in{R}^{I}$\
> > **Outputs.** Conflict (CF) and ignorance (IG)\
> > *Note: Equation numbering follows the main text.*
> > >1. Estimate belief-assignment parameters via LCP
> > $$A = W - \mu_0(W), B=-\bigl(A-\mu_1(A)\bigr)\odot Z^\top,\tag{8}$$
> > where $\mu_0,\mu_1$ are centering operators (e.g., mean), and $\odot$ denotes elementwise product.
> > 2. Compute evidence weight matrix ${E}\in{R}^{I\times J}=${$e_{ij}$}
> > $$E=A\odot Z^\top+B.\tag{5}$$
> > 3. Separate positive and negative evidence (see line 231 in the main text)
> > $$E^+=\max(0,E),{E}^-=\max(0,-E).$$
> > 4. Model positive / negative simple basic belief assignments
> > $$m_{ij}^+(\{h_j\})=1-\exp\bigl(-e_{ij}^+\bigr),m_{ij}^-\bigl(\overline{\{h_j\}}\bigr) = 1 - \exp\bigl(-e_{ij}^-\bigr).\tag{6}$$
> > 5. Fuse simple basic belief assignments
> > $$e_{j}^+=\sum_i e_{ij}^+, e_{j}^-=\sum_i e_{ij}^-,\tag{9}$$
> > $$m_j^+(\{h_j\}) = 1 - \exp\bigl(-e_j^+\bigr),m_j^-\bigl(\overline{\{h_j\}}\bigr)=1-\exp\bigl(-e_j^-\bigr).\tag{10}$$
> > 6. Compute global positive / negative basic belief assignments (see line 270 in the main text)
> > $$m^+=\bigoplus_j m_j^+,m^-=\bigoplus_j m_j^-.$$
> > 7. Compute conflict (CF) and ignorance (IG)
> > $$CF=\sum_{\mathcal{S}_1\cap\mathcal{S}_2=\varnothing}m^+(\mathcal{S}_1)\cdot m^-(\mathcal{S}_2),IG=\sum_j m_j^-(\mathcal{H}).\tag{11}$$

---

> > > ### Author Response · Authors · 2025-11-22
> > >
> > > **Q1: Method motivation and novelty (Weakness 1)**
> > > >**Reply**: Prior methods typically use measures such as entropy, log-likelihood, or variance to represent uncertainty, but these metrics cannot reveal the diverse sources behind model errors. As shown in our key observation (Figure 1 in the main paper), LVLM misbehaviors could stem from distinct sources, such as conflicts between visual evidence and the model’s knowledge gaps.
> > > This motivates our design: EUQ leverages decoder-head features to separately quantify conflict (inconsistency among internal representations) and ignorance (insufficient reliable evidence).
> > > This separation is a novel aspect of LVLM uncertainty modeling, enabling EUQ to differentiate among various misbehavior types.
> > > Importantly, our method is not only effective but also efficient, achieving a **15%** improvement over the strongest baseline and running $\boldsymbol{10^3}$ times faster than multi-sampling baselines (except PE and LN-PE).
> > >
> > > **Q2: Method applicability (Weakness 1 & 2)**
> > > >**Applicability beyond VLMs**\
> > > EUQ is not tied to LVLM-specific components.
> > > The method only requires access to the projection (output) layer, an architectural component that also exists in models such as BERT, ResNet, and LLMs. Thus, EUQ can be applied to a broad range of neural networks. To illustrate this applicability, Table 10 (Appendix A7.3) provides a toy example on convolution neural networks, demonstrating that EUQ easily extends beyond VLMs.\
> > > **Applicability to closed-source model**\
> > > As discussed in Section~6 (DISCUSSION), the requirement to access internal representations restricts the applicability of EUQ to closed-source APIs. However, our primary goal is to provide model developers with trustworthy, interpretable, and fine-grained signals about model uncertainty, which can help identify potential misbehavior, guide safe deployment, and inform model development and debugging.
> > > Access to decoder-layer activations enables EUQ to separate conflict from ignorance, which output-only methods cannot achieve.
> > >
> > > **Q3: Baseline selection (Weakness 3)**
> > > >**Reply**: Semantic entropy (SE) [1] remains applicable even when the target output is a single word. Multiple sampled outputs may still appear in diverse surface forms (e.g., ``A``, ``A (option)``, ``a``, ``I think the answer is A``) especially under high temperature, since LVLMs do not strictly follow the prompt. Clustering is therefore necessary to group semantically equivalent responses before computing uncertainty.
> > > In cases where all generations collapse to a single canonical form, SE naturally degradate to self-consistency (SC) [2]. This matches our experimental observation that SE and SC perform similarly in such settings.
> > >
> > > **Q4: Typo near line 393 (Question 1)**
> > > >**Reply**: In the current version (Table 4 in the main text), the rows correspond to *models and metrics*, while the columns correspond to *methods*. This will be modified in the revised version.
> > >
> > > **Q5: Correctness Metric (Question 2)**
> > > >**Reply**: For our tasks, we use ROUGE-L (threshold > 0.5) to determine whether a model's output is correct. This is necessary because, especially under high temperature, models often do not strictly follow the prompt format, making direct string matching unreliable. Instead, we compute the ROUGE-L score between the model’s response (e.g., ``The answer is A. (option)``) and the correct option (``A``); if the score exceeds 0.5, the answer is considered correct. This approach effectively measures accuracy while being more robust than direct string matching.
> > >
> > > **Q6: Prompt Design (Question 3)**
> > > >**Reply**: Our experiments focus on detecting misbehaviors (e.g., hallucinations) rather than mitigating them. While reviewers mention that having the model reject uncertain inputs is a valid mitigation strategy, our image-text pairs are intentionally designed to induce misbehaviors, which are necessary for evaluating detection performance.
> > >
> > > **Reference**
> > > >[1] Kuhn L, Gal Y, Farquhar S. Semantic Uncertainty: Linguistic Invariances for Uncertainty Estimation in Natural Language Generation[C]//ICLR 2023.\
> > > [2] Wang X, et al. Self-Consistency Improves Chain of Thought Reasoning in Language Models[C]//ICLR 2022.

---

> > > > ### Comment · Reviewer_iGLE · 2025-11-22
> > > >
> > > > I would like to thank the author for the response and the
> > > >
> > > >
> > > > ## Method applicability
> > > >
> > > > Since the method itself is not only applicable to VLM, do the issues of
> > > >
> > > > - quantify conflict (inconsistency among internal representations)
> > > > - ignorance (insufficient reliable evidence)
> > > >
> > > > also applicable to other models?
> > > >
> > > > Following this, can the method be used for uncertainty quantification? I believe Appendix A.7.3 verifies this.
> > > >
> > > > Then I guess the method is certainly applicable to language modality hallucination detection.
> > > >
> > > > Based on these results, I would suggest the authors broaden the scope of this paper and frame this as a generic approach to all types of settings and models.
> > > >
> > > > ##  Correctness Metric
> > > >
> > > > I still don't understand it, what if the model's response is:
> > > > ```
> > > > Answer: (B)
> > > > ```
> > > > and the ground truth answer is A, then what would the Rouge score be?
> > > >
> > > > Can the author just use e.g. LLM as a judge, or use regex to extract the answer?
> > > >
> > > > ## Re Prompt Design
> > > >
> > > > My argument here is that, the prompt used in the experiments is forcing the model to misbehave or hallucinate: If you *force* the model to pick between some non-sensible options, of course, the model will hallucinate. However modern VLMs are clever, they are not ResNet, you can tweak the prompt (by e.g. including `None of above` option) and newer VLMs can do a fairly good job at detecting abnormal inputs.
> > > >
> > > >
> > > > -----------------
> > > >
> > > > I will consider modifying my score after discussion with other reviewers. Again, this is my first time learning about Theory Of Evidence, so my judgment may be imprecise and I may underestimate the value and the contribution. (implied by my confidence score of 2)

---

> > > > > ### Author Response · Authors · 2025-11-27
> > > > >
> > > > > We sincerely thank the reviewer for the timely and constructive feedback. We have provided responses to each of your concerns, and we are happy to provide further clarification if needed.\
> > > > > **Method applicability**
> > > > > > **Reply:** We agree that our method is grounded in a general uncertainty quantification (UQ) framework. As you mention the verification in Appendix A.7.3, the method effectively handles OOD and adversarial detection on smaller CNN-based architectures.\
> > > > > However, we chose to focus this paper on LVLMs to address a critical gap in the current landscape. While UQ research remains an active research area, established methods such as MC Dropout [1], Deep Ensembles [2], EDL [3], ENN [4], ABNN [5], and RS-NN [6] primarily excel in small-scale settings. These methods often struggle with **scalability** when applied to massive foundation models due to prohibitive computational costs. This gap motivates our design of an evidence-based UQ method for addressing the urgent safety challenges unique to LVLMs, such as hallucinations and jailbreaks. Crucially, our method goes beyond total uncertainty by explicitly distinguishing between **conflict** (conflicting internal knowledge) and **ignorance** (lack of effective information). Characterizing these two distinct sources provides a valuable fine-grained perspective, allowing us to capture distinct distributional patterns associated with different LVLM misbehaviors.\
> > > > > We find your suggestion to broaden the scope truly encouraging. We plan to initiate a comprehensive benchmarking effort to evaluate our proposed method alongside existing UQ works across a wider range of models and learning tasks in future work.
> > > > >
> > > > > **Correctness metric**
> > > > > >**Reply:** We thank the reviewer for the opportunity to further clarify the correctness metric.
> > > > > The correctness is determined by the ROUGE score between the ground-truth string and the model’s generated answer. For example:
> > > > > Ground truth: **"A. yes"**
> > > > >
> > > > > | Candidate answer| ROUGE | Judgement|
> > > > > |-|-|-|
> > > > > | a. yes| 1.00| correct|
> > > > > | answer: a. yes| 0.80| correct|
> > > > > | A| 0.67| correct|
> > > > > | a| 0.67| correct|
> > > > > | answer: A.| 0.50| correct|
> > > > > | B. no| 0.00| hallucinated|
> > > > > | B| 0.00| hallucinated|
> > > > > | answer: B| 0.00| hallucinated|
> > > > > | answer: b. no| 0.00| hallucinated|
> > > > >
> > > > > >These examples demonstrate that ROUGE offers a consistent and reliable way to assess correctness across different answer formats. Using ROUGE for correctness determination is also a widely adopted convention in prior work [7,8].\
> > > > > We agree that alternatives such as LLM-based judging or regex-based answer extraction are feasible. We opted for ROUGE just to maintain a deterministic, reproducible evaluation free from additional dependencies.
> > > > >
> > > > >
> > > > > **Prompt design**
> > > > > >**Reply:** Thank you very much for raising this concern again. We realize there may have been a misunderstanding. All provided options are reasonable, and each question includes a valid correct answer.\
> > > > > Nevertheless, we appreciate the reviewer’s insightful suggestion, and we also explored this idea by adding a “None of the above” option, following the idea in [9]. The models almost never selected it across 1,500 samples, Qwen2.5-VL selected it **4 times** and InternVL2.5 **0 times**. We then further strengthened the prompt: ``If you are unsure, please select “None of the above”.``Even with this instruction, the models rarely chose the option (Qwen2.5-VL: **74**; InternVL2.5: **8**), indicating persistent overconfidence.\
> > > > > These results suggest that modern LVLMs often remain overconfident, and explicit “I don’t know” training (e.g., [10]) is one potential way to mitigate this. We will incorporate these discussions and the insights from [9] into the revised version.
> > > > >
> > > > > **Reference**
> > > > > >[1] Srivastava N, Hinton G, Krizhevsky A, et al. Dropout: a simple way to prevent neural networks from overfitting[J]. JMLR 2014.\
> > > > > [2] Lakshminarayanan B, Pritzel A, Blundell C. Simple and scalable predictive uncertainty estimation using deep ensembles[J]. NeurIPS 2017.\
> > > > > [3] Sensoy M, Kaplan L, Kandemir M. Evidential deep learning to quantify classification uncertainty[J]. NeurIPS 2018.\
> > > > > [4] Osband I, Wen Z, Asghari S M, et al. Epistemic neural networks[J]. NeurIPS 2023.\
> > > > > [5] Franchi G, Laurent O, Leguéry M, et al. Make me a bnn: A simple strategy for estimating bayesian uncertainty from pre-trained models[C]//CVPR 2024.\
> > > > > [6] Manchingal S K, Mubashar M, Wang K, et al. Random-set neural networks[C]//ICLR 2025.\
> > > > > [7] Kuhn L, Gal Y, Farquhar S. Semantic Uncertainty: Linguistic Invariances for Uncertainty Estimation in Natural Language Generation[C]//ICLR 2023.\
> > > > > [8] Wang X, et al. Self-Consistency Improves Chain of Thought Reasoning in Language Models[C]//ICLR 2022.\
> > > > > [9] Wang X, Nalisnick E. Are vision language models robust to uncertain inputs?// arXiv 2025.\
> > > > > [10] Zhang H, Diao S, Lin Y, et al. R-tuning: Instructing large language models to say ‘i don’t know’[C]//NAACL 2024.

---

> > > > > > ### Author Response · Authors · 2025-12-01
> > > > > > **Summary of Rebuttal for Reviewer iGLE**
> > > > > >
> > > > > > # Summary of Rebuttal for Reviewer iGLE
> > > > > >
> > > > > > Reviewer iGLE raised concerns on method clarity, applicability, correctness metric, and prompt design. We addressed them as follows:
> > > > > >
> > > > > > 1. **Method Clarity and DST Concepts**
> > > > > > We added a clear explanation of Dempster–Shafer Theory, introduced a light-bulb–switch toy example to illustrate the DST Concepts, and included a concise algorithm block summarizing the computational steps of EUQ. **We also moved extended explanations and additional examples to Appendix A.3** to make the main pipeline easier to follow.
> > > > > > 2. **Method Motivation and Novelty**
> > > > > > Existing metrics like entropy cannot distinguish diverse error sources. EUQ leverages decoder-head features to separately quantify *conflict (inconsistency among internal representations) and ignorance (insufficient reliable evidence), enabling it to differentiate misbehavior types. The method is both effective, achieving a **15%** improvement over the strongest baseline, and highly efficient, running $\boldsymbol{10^3}$ times faster than multi-sampling baselines.
> > > > > >
> > > > > > 3. **Applicability**
> > > > > > Experimental validation on smaller CNN architectures was already provided in Appendix A.7.3. We clarify that EUQ is generally applicable beyond LVLMs, including architectures such as BERT, ResNet, and LLMs, **as discussed in the updated Section 6 of the revised version.**
> > > > > >
> > > > > > 4. **Correctness Metric**
> > > > > >    We clarified that ROUGE-L is used for robust and reproducible correctness evaluation across diverse answer formats, addressing potential issues with string matching.
> > > > > >
> > > > > >
> > > > > > 5. **Prompt Design**
> > > > > > We tested a “None of the above” option. Models rarely selected it, confirming persistent overconfidence and demonstrating the necessity of EUQ for misbehavior detection. **The corresponding analysis and results have been included in the revised version.**
> > > > > >
> > > > > >
> > > > > > We have addressed all concerns raised by Reviewer iGLE with clarifications and additional validation, demonstrating EUQ’s significance, novelty, and broad applicability.\
> > > > > > As the reviewer noted, ***“I will consider modifying my score after discussion with other reviewers” and acknowledged that they may underestimate the value and contribution of our work.***

---

### Author Response · Authors · 2025-12-02
**Summary Comment**

Dear Area Chairs, Senior Area Chairs, and Program Chairs,

We appreciate your efforts in taking over our submission under the special circumstances caused by the recently disclosed OpenReview bug. We fully understand and support the decisions made by the ICLR organizers, including reassigning each paper to a new area chair. Therefore, we provide this brief summary comment to highlight our contributions and how our rebuttal addresses the reviewers’ concerns, in support of your independent evaluation.

**Contribution**
>Large vision-language models (LVLMs) are not fully reliable and can exhibit various misbehaviors, such as hallucinations or jailbreaks. Our key observation (Figure 1 in the main paper) shows that these misbehaviors can stem from distinct sources of uncertainties, including conflicts between visual evidence and the model’s knowledge gaps. We introduce a unified evidential perspective that quantifies distinct uncertainty types to effectively detect diverse misbehaviors, providing a principled solution for improving LVLM reliability.\
1、We identify two types of epistemic uncertainties underlying LVLM misbehaviors and propose a computationally efficient DST-based detection method (**EUQ**) that leverages conflict (**CF**) and ignorance (**IG**) to capture them in a single forward pass.\
2、We perform a **layer-wise dynamic analysis** of internal representations, enabling certain layers to distinguish among all four misbehavior categories.\
**Experimental Validation**: Extensive experiments on four LVLMs across four misbehaviors show that EUQ improves the strongest baseline by **15%** while running $\boldsymbol{10^3}$ times faster than multi-sampling methods.

**Reviewer Highlights**
>We sincerely thank all reviewers for their time and thoughtful evaluations, and we are especially appreciative of their recognition of the **novelty and value of our work** (TyT1) and noting that it is an **interesting study** (kLDs). Reviewers praised EUQ’s **mathematical rigor** (IGLE), **computational efficiency** (TyT1, RBSV), and the effectiveness of CF and IG across models and tasks, with **superior preformace compared to baselines** (all reviewers). They also highlighted the **clear writing** (TyT1, RBSV), **insightful layer-wise uncertainty analysis** (TyT1, kLDs), and noted that our experiments cover a **wide range of tasks, datasets, and models** (IGLE, TyT1, RBSV).\
**Overall, reviewers acknowledged that the method is promising, effective, and broadly applicable, with multiple reviewers expressing that they would not oppose acceptance.**

**Reviewers’ Concerns and Our Responses**
>1、Method Clarity and DST Concepts (iGLE)\
We **added a concise introduction to Dempster–Shafer Theory, a simple illustrative toy example, and a compact algorithm summarizing the EUQ computation**. We also moved extended explanations and additional examples to **Appendix A.3** to make the main pipeline of the proposed method easier to follow.\
2、Applicability of EUQ (iGLE)\
Experimental validation for this point was **already included in Appendix A.7.3 prior to the rebuttal**. We **further clarify in Section 6** that EUQ is applicable to architectures such as BERT, ResNet, and LLMs, extending applicability beyond LVLMs.\
3、Interpretability of CF/IG and Semantic Explanation (TyT1, kLDs)\
We computed CF and IG on the CoT example in Figure 1, finding that CF highlights objects (e.g., diaper) while IG also captures modifiers (e.g., vaguely), showing EUQ’s ability to localize uncertainty in reasoning traces.
We have **improved the corresponding text–heatmap visualizations in Figure 1** of the revised version, following the constructive suggestion from reviewer TyT1.\
4、Baselines, Dataset, and Model Scale (kLDs,RBSV)\
Our evaluation covers **four misbehavior types, 11 datasets, 29k test cases, and four LVLMs**, which reviewers (iGLE, TyT1) recognized as extensive. We also added **HiddenDetect (Jiang et al., 2025)** as a baseline and **Qwen2.5-VL-72B (Bai et al., 2025)** experiments to strengthen comparisons and large-model validation.\
5、Prompt Design and Evaluation Protocol (RBSV, iGLE)\
We explored adding a “None of the above” option, **following the approach in (Wang & Nalisnick, 2025)**, and included the results in **Section 5.2**. The models rarely selected this option, highlighting the necessity of methods like EUQ.


**Reviewer Feedback**
>Due to the recent OpenReview bug, only Reviewer iGLE provided a response. The reviewer indicated that they may consider modifying their score after discussion and acknowledged that they may have underestimated the value and contributions of our work.


For a detailed point-by-point response, please refer to our rebuttal for each reviewer.\
We have ensured that EUQ poses no safety or fairness risks and complies with ICLR’s ethical guidelines.

Thank you very much again for your time and consideration.\
Warm regards,\
The Authors of Paper 15301

---

### Meta-Review · Area_Chair_YuZD · 2025-12-04

**Summary:**

The paper received initial scores of 6, 4, 8, and 4, which is in the borderline range for acceptance. Reviewers TyT1 and RBSV found the proposed hallucination-detection method interesting and novel, while reviewers kLDs and iGLE raised concerns regarding scalability, baseline selection, and the interpretation of the evaluation metrics. After carefully reviewing the comments and the authors’ responses, I find that nearly all concerns have been adequately addressed. Therefore, I believe the paper meets the acceptance criteria for ICLR.

**Reviewer Concerns:**

Nearly all concerns raised by the reviewers (iGLE, TyT1, kLDs, and RBSV) have been adequately addressed.

**Reviewer Scores:**

I believe that if the reviewer iGLE had participated in the discussion phase, the score would likely have been updated upward.

---

### Decision · Program_Chairs · 2026-01-26

Accept (Poster)